

# Bridging Time and Frequency: A Joint Modeling Framework for Irregular Multivariate Time Series Forecasting

**Xiangfei Qiu** [1]  **Kangjia Yan** [1]  **Xvyuan Liu** [1]  **Xingjian Wu** [1]  **Jilin Hu** [1 2 3]

## Abstract

Irregular multivariate time series forecasting (IMTSF) is challenging due to non-uniform sampling and variable asynchronicity. These irregularities violate the equidistant assumptions of standard models, hindering local temporal modeling and rendering classical frequency-domain methods ineffective for capturing global periodic structures. To address this challenge, we propose TFMixer, a joint time–frequency modeling framework for IMTS forecasting. Specifically, TFMixer incorporates a Global Frequency Module that employs a learnable Non-Uniform Discrete Fourier Transform (NUDFT) to directly extract spectral representations from irregular timestamps. In parallel, the Local Time Module introduces a query-based patch mixing mechanism to adaptively aggregate informative temporal patches and alleviate information density imbalance. Finally, TFMixer fuses the time-domain and frequency-domain representations to generate forecasts and further leverages inverse NUDFT for explicit seasonal extrapolation. Extensive experiments on real-world datasets demonstrate the state-of-the-art performance of TFMixer.

## 1. Introduction

Irregular Multivariate Time Series (IMTS) are defined as collections of univariate variables where observations are recorded at non-uniform temporal intervals and exhibit inter-variable asynchronicity (Kidger et al., 2020; Liu et al., 2026b; Qiu et al., 2025c). Unlike traditional regular time series, IMTS do not adhere to a fixed sampling rate, often resulting in missing values and sparse data segments across

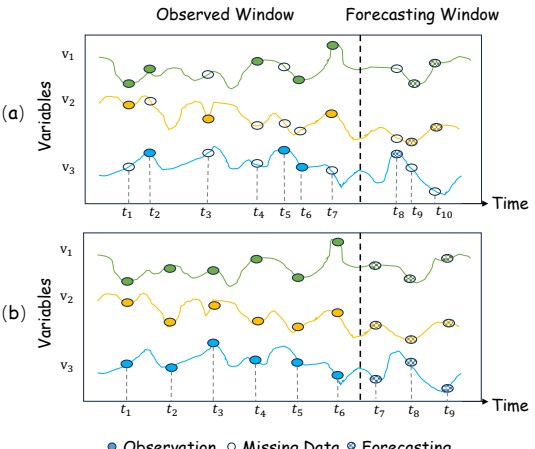

*Figure 1.* The comparison of irregular multivariate time series forecasting (a) and regular multivariate time series forecasting (b).

different channels—see Figure 1.

The importance of IMTS modeling is paramount in numerous real-world applications where data is naturally irregular (Yao et al., 2018; Brouwer et al., 2019; Wu et al., 2025c; Gao et al., 2025). In healthcare, patient vitals are recorded at varying frequencies based on clinical needs; in environmental monitoring, sensor failures or power-saving modes lead to sporadic data streams; and in finance, trade events occur at irregular ticks. Accurate forecasting in these domains is critical for proactive decision-making and system stability.

Irregular Multivariate Time Series Forecasting (IMTSF) is inherently challenging due to the intricate interplay between long- and short-range temporal dependencies (Luo et al., 2025; Liu et al., 2026b). Long-range (global) dependencies typically manifest as seasonal or periodic patterns spanning the entire historical horizon, whereas short-range (local) dependencies represent fine-grained dynamics occurring within localized intervals—see Figure 2. Successfully capturing both scales is critical for accurate forecasting in irregular settings.

From a modeling perspective, frequency-domain methods provide a natural and effective mechanism for capturing long-term periodic structures and global dependencies.

---

[1]School of Data Science and Engineering, ECNU [2]Shanghai Engineering Research Center of Big Data Management [3]National Institutes of Educational Policy Research, ECNU. Correspondence to: Jilin Hu <jlhu@dase.ecnu.edu.cn>.

*Proceedings of the 43rd International Conference on Machine Learning*, Seoul, South Korea. PMLR 306, 2026. Copyright 2026 by the author(s).

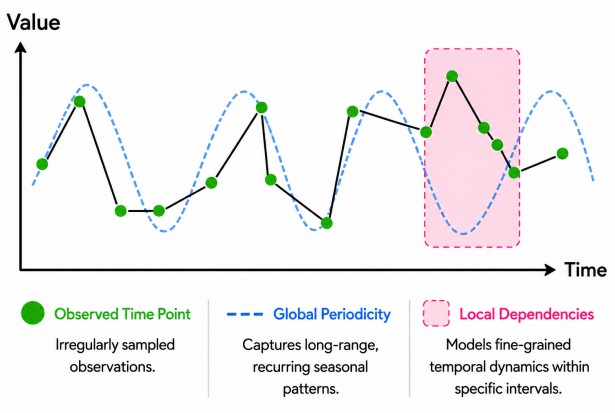

*Figure 2.* Global and local temporal dependencies in IMTSF. The observed time points (green dots) are sampled non-uniformly, creating challenges for traditional models. The global periodicity (blue dashed line) represents long-range seasonal patterns, while local dependencies (pink shaded area) capture fine-grained temporal dynamics within specific intervals.

Fourier-based techniques and their neural extensions have demonstrated strong performance on regularly sampled time series (Zhou et al., 2022b; Wu et al., 2023). However, classical spectral operators such as the Discrete Fourier Transform (DFT) and the Fast Fourier Transform (FFT) fundamentally rely on the assumption of uniform sampling, which rarely holds for irregular time grids, thereby limiting their direct applicability to IMTSF.

At the same time, irregular sampling substantially complicates local temporal representation learning. Many IMTSF approaches segment observations into local temporal patches to extract fine-grained dynamics (Zhang et al., 2024b; Liu et al., 2025; Luo et al., 2025). Under non-uniform sampling, the imbalance in information density presents a significant challenge. These patches often exhibit highly uneven distributions: while some contain dense observations with rich temporal structures, others are sparse or entirely vacant. This imbalance makes reliable feature aggregation particularly challenging and, if not explicitly addressed, can introduce noise and significantly degrade representation quality.

Taken together, these limitations expose a fundamental challenge in IMTS forecasting: *how to jointly model global frequency structures and local temporal dynamics under irregular sampling*. Addressing this challenge requires not only frequency-domain operators that can directly operate on irregular timestamps, but also time-domain mechanisms capable of robustly aggregating information from unevenly distributed observations.

To address these challenges, we propose TFMixer, a joint modeling framework that bridges time and frequency for IMTS forecasting. TFMixer explicitly decouples global pe-

riodic modeling from local temporal representation learning and processes them in parallel. Specifically, the *Global Frequency Module* employs a learnable Non-Uniform Discrete Fourier Transform (NUDFT) to directly extract spectral representations from irregularly sampled time series, avoiding the need for interpolation. The resulting spectral coefficients are further processed by a spectrum encoder to capture global frequency-domain features. In parallel, the *Local Time Module* introduces a query-based patch mixing mechanism, in which a fixed set of learnable query tokens selectively attends to informative patches. This design alleviates information density imbalance and yields a compact local temporal representation. Finally, the *Output Module* fuses the complementary time-domain and frequency-domain representations to generate predictions and further leverages the refined spectral coefficients to perform inverse NUDFT for explicit seasonal extrapolation.

Our contributions are summarized as follows:

- We propose TFMixer, a unified joint time–frequency framework for irregular multivariate time series forecasting that explicitly bridges global spectral dependencies and local temporal dynamics.

- We introduce a learnable NUDFT that directly operates on irregular timestamps. This approach facilitates the extraction of frequency-domain representations and the generation of additional seasonal biases.

- We identify the information density imbalance problem induced by patch-based approaches in IMTSF and propose a novel query-based patch mixing mechanism to selectively aggregate local temporal representations.

- Extensive experiments on multiple real-world IMTSF datasets demonstrate that TFMixer achieves state-of-the-art performance. Additionally, all datasets and code are available at: https://github.com/decisionintelligence/TFMixer.

## 2. Related works

### 2.1. Irregular Multivariate Time Series Forecasting

IMTSF is a critical task in domains such as clinical medicine and meteorology (Zhang et al., 2023; Li et al., 2025; Liu et al., 2026b). Early research primarily focused on using Recurrent Neural Networks (RNNs) modified with temporal decay components or missing value indicators to handle irregular gaps, such as the GRU-D model (Che et al., 2016). Another prominent line of work employs Neural Ordinary Differential Equations (Neural ODEs) to model the continuous-time evolution of hidden states between observations (Chen et al., 2018). Recently, the field has shifted toward more flexible architectures. Set-based models like

SeFT (Horn et al., 2020) treat observations as unordered sets to eliminate the need for fixed time grids. Graph-based methods, such as Raindrop (Zhang et al., 2022) and ASTGI (Liu et al., 2026a), have been developed to capture the complex relationships between multiple sensors with asynchronous sampling. More recently, patching mechanisms have been adapted for IMTSF. For example, tPatchGNN (Zhang et al., 2024b) segments irregular sequences into transformable patches to capture local structures. However, these methods often struggle with information density imbalances across patches and primarily operate in the time domain, missing frequency domain analysis.

### 2.2. Frequency Domain Analysis in Time Series

Frequency domain analysis is highly effective for capturing long-term periodic patterns, which are often obscured in the time domain. For regularly sampled data, the Fast Fourier Transform (FFT) is the standard tool. FEDformer (Zhou et al., 2022b) incorporates a frequency-enhanced Transformer with a seasonal-trend decomposition to improve long-term forecasting. FiLM (Zhou et al., 2022a) applies Legendre Polynomials and Fourier projections to preserve structural information and remove noise. TimesNet (Wu et al., 2023) further advances this by transforming 1D time series into 2D tensors based on multiple periods, applying 2D kernels to extract intra-period and inter-period variations. However, applying frequency analysis to irregular multivariate time series is challenging because FFT requires uniform grids. To address this, KAFNet (Zhou et al., 2026) utilizes Canonical Pre-Alignment to project irregular observations onto a shared temporal grid, allowing for global inter-series correlation modeling in the frequency domain. And it maps the aligned sequences into a fixed-dimensional latent space, where spectral analysis is performed on the hidden features rather than the raw observations. Unlike previous approaches, TFMixer integrates a learnable NUDFT module, enabling the direct extraction of spectral features from datasets with raw non-uniform observations.

## 3. Preliminary

### 3.1. Problem Definition

An Irregular Multivariate Time Series (IMTS) $\mathcal{O}$ is defined as a collection of $N$ univariate variable $\{o_{1:L_n}^n\}_{n=1}^N$. Each variable $o_{1:L_n}^n$ consists of $L_n$ observation triplets $(t_i^n, x_i^n)$, where $t_i^n$ denotes the timestamp and $x_i^n$ represents the corresponding observed value for the $i$-th observation of the $n$-th variable. These variables are characterized by non-uniform temporal intervals and inter-variable asynchronicity.

The IMTSF task is formulated as follows: given the historical observations $\mathcal{O}$ and a set of target prediction queries $\mathcal{Q} = \{[q_j^n]_{j=1}^{Q_n}\}_{n=1}^N$ (where each $q_j^n$ represents a future timestamp for variable $n$), the goal is to develop and optimize a predictive model $\mathcal{F}(\cdot)$ that generates accurate estimates of future values $\hat{\mathcal{V}} = \{[\hat{v}_j^n]_{j=1}^{Q_n}\}_{n=1}^N$. This mapping can be expressed as:

$$\hat{\mathcal{V}} = \mathcal{F}(\mathcal{O}, \mathcal{Q}). \tag{1}$$

### 3.2. Canonical Pre-Alignment Representation for IMTS

To facilitate IMTS modeling, we adopt a pre-alignment representation method (Zhang et al., 2024b; Che et al., 2018), which has been widely in current researches (Brouwer et al., 2019; Zhang et al., 2023; Schirmer et al., 2022; Mercatali et al., 2024). In this method, an IMTS $\mathcal{O}$ is represented by three matrix $(\mathcal{T}, \mathcal{X}, \mathcal{M})$. $\mathcal{T} = [t_l]_{l=1}^L = \bigcup_{n=1}^N [t_i^n]_{i=1}^{L_n} \in \mathbb{R}^L$ denotes the chronological unique timestamps of all observations within $\mathcal{O}$. $\mathcal{X} = [[\tilde{x}_l^n]_{n=1}^N]_{l=1}^L \in \mathbb{R}^{L \times N}$ are variable's values corresponding to the timestamps, where $\tilde{x}_l^n = x_i^n$ if the value of $n$-th variable is observed at time $t_l$, otherwise $\tilde{x}_l^n$ would be filled zero. $\mathcal{M} = [[m_l^n]_{n=1}^N]_{l=1}^L \in \mathbb{R}^{L \times N}$ represents a masking matrix, where $m_l^n = 1$ if $\tilde{x}_l^n$ is observed at time $t_l$, otherwise zero.

## 4. Methodology

### 4.1. Structure Overview

Figure 3 illustrates the overall architecture of TFMixer. Specifically, we first introduce a *Masked RevIN Module* to apply masked normalization to the raw input. Subsequently, we design the *Global Frequency Module*, which utilizes a Learnable NUDFT to map signals into the frequency domain, where a spectrum encoder extracts global frequency features. Concurrently, we propose the *Local Time Module*, which begins with patch partitioning and initial encoding. To address the non-uniform information density of patches in irregular time series, we introduce a Query-based Patch Attention mechanism and Dual-Mixing Blocks to deeply explore temporal dependencies between patches and inter-variable correlations. Following this, the *Output Module* facilitates the joint fusion of time and frequency features, employing an MLP decoder to generate the initial predictions. The final prediction is synthesized by combining the initial predictions with the seasonal bias derived from future timestamps via Inverse NUDFT, followed by a Masked Denormalization step.

Below, we will introduce the details of each module in our framework. Technical details regarding the Masked RevIN Module are provided in the Appendix C.1.

### 4.2. Global Frequency Module

The Global Frequency Module is designed to capture overarching periodic dependencies that are often obscured by irregular sampling and missing values in the time domain.

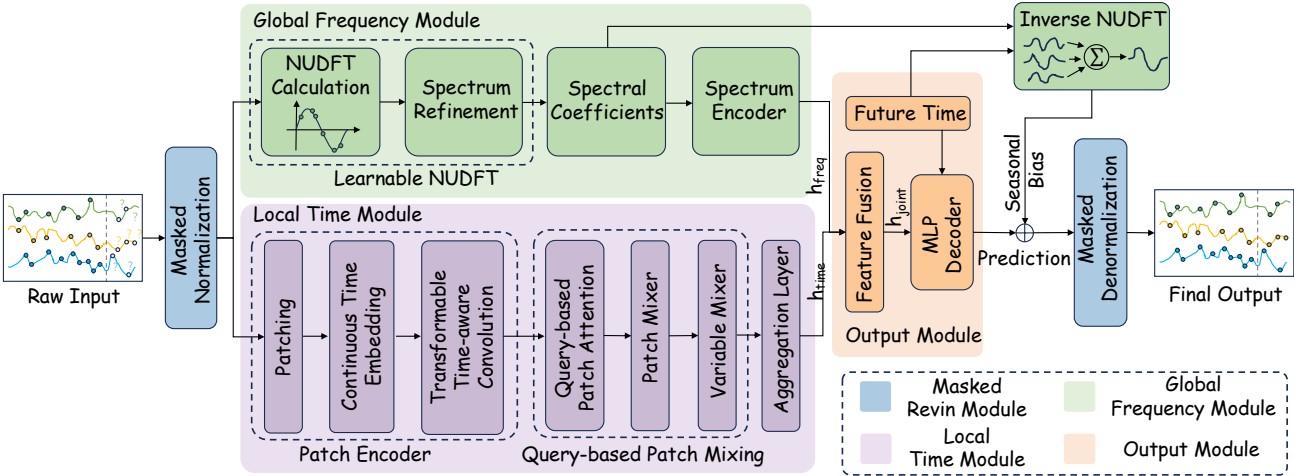

*Figure 3.* TFMixer architecture. (1) Masked RevIN Module normalizes/denormalizes the data. (2) Global Frequency Module captures long-range periodic dependencies through a learnable Non-Uniform Discrete Fourier Transform (NUDFT). (3) Local Time Module extracts fine-grained patterns using a transformable patch encoder followed by query-based attention and dual-mixing blocks. (4) Output Module performs joint time-frequency feature fusion to generate the final forecasting results.

Unlike the standard Fast Fourier Transform (FFT) which requires inputs to be on a regular grid, we implement a Learnable Non-Uniform Discrete Fourier Transform (NUDFT) to directly process the irregular triplets $(\mathcal{T}, \mathcal{V}, \mathcal{M})$.

### 4.2.1. LEARNABLE NUDFT

**NUDFT Calculation.** Given a set of learnable frequencies $\{\omega_k\}_{k=1}^{K}$, we project the normalized observed values $\mathcal{V}$ onto a set of non-uniform basis functions. For each variable $n$, the real and imaginary components of the spectrum at frequency $\omega_k$ are calculated as:

$$R_n(\omega_k) = \frac{1}{Z_n} \sum_{l=1}^{L} m_l^n v_l^n \cos(2\pi\omega_k t_l) \in \mathbb{R}^{N \times K}, \quad (2)$$

$$I_n(\omega_k) = -\frac{1}{Z_n} \sum_{l=1}^{L} m_l^n v_l^n \sin(2\pi\omega_k t_l) \in \mathbb{R}^{N \times K}, \quad (3)$$

$$Z_n = \max\left(\sum_{l=1}^{L} m_l^n, \epsilon\right), \quad (4)$$

where $Z_n$ is a normalization factor representing the count of valid observations for variable $n$, and $\epsilon$ is a small constant to prevent division by zero. We then construct the raw spectral representation $\mathbf{S}_{\text{raw}} \in \mathbb{R}^{N \times 2K}$ by concatenating the real and imaginary parts:

$$\mathbf{S}_{\text{raw}} = [R_n(\omega_k) \parallel I_n(\omega_k)] \in \mathbb{R}^{N \times 2K}, \quad (5)$$

where $\parallel$ denotes the concatenation operation.

**Spectrum Refinement.** To compensate for the limited expressiveness of the purely linear NUDFT calculation, we introduce a refinement stage within the learnable NUDFT

block. We employ a Multi-Layer Perceptron (MLP) to process the raw coefficients:

$$\mathbf{S}_{\text{refined}} = \text{MLP}_{\text{refine}}(\mathbf{S}_{\text{raw}}) = [\hat{R}_n(\omega_k) \parallel \hat{I}_n(\omega_k)] \in \mathbb{R}^{N \times 2K}, \quad (6)$$

where $\text{MLP}_{\text{refine}}(\cdot)$ applies non-linear transformations while preserving the original spectral dimensionality $N \times 2K$. This step allows the model to adaptively refine the amplitude and phase information, capturing complex, non-stationary frequency patterns that are difficult to extract through analytical derivation alone. This refined spectrum $\mathbf{S}_{\text{refined}} = [\hat{R}_n(\omega_k) \parallel \hat{I}_n(\omega_k)]$ provides the final optimized Spectral coefficients for the subsequent Inverse NUDFT, ensuring that the generated seasonal bias accurately reflects complex periodicities.

### 4.2.2. SPECTRUM ENCODER

After obtaining the refined spectrum $\mathbf{S}_{\text{refined}}$, the Spectrum Encoder is responsible for extracting high-level frequency features and projecting them into the model's latent space to facilitate joint modeling with the local time branch. We project the refined spectral features into the hidden dimension $D$ using a dedicated projection layer followed by normalization:

$$\mathbf{h}_{\text{freq}} = \text{LayerNorm}\left(\text{MLP}_{\text{enc}}(\mathbf{S}_{\text{refined}})\right), \quad (7)$$

where $\mathbf{h}_{\text{freq}} \in \mathbb{R}^{N \times D}$ serves as the global frequency-domain representation.

In summary, the Global Frequency Module outputs two key components: (1) the latent representation $\mathbf{h}_{\text{freq}}$ for feature fusion in the Output Module, and (2) the refined spectral coefficients $\mathbf{S}_{\text{refined}}$ (which retains the $N \times 2K$ structure) for subsequent Inverse NUDFT to provide a seasonal bias.

## 4.3. Local Time Module

The Local Time Module is engineered to capture fine-grained temporal patterns and inter-variable correlations from irregular observations. By leveraging a patch encoder mechanism and the query-based patch mixing mechanism, the module adaptively handles the non-uniform information density inherent in IMTSF.

### 4.3.1. PATCH ENCODER

**Patching.** Time series patching has demonstrated significant efficacy in capturing local semantic information while reducing computational complexity. Unlike standard patching for regular time series that segments data into a fixed number of observations, we adopt the *transformable patching* strategy inspired by tPatchGNN (Zhang et al., 2024b) to handle the inherent irregularity of IMTS. Formally, given a univariate irregular time series $\mathbf{o}_{1:L}$, we partition it into a sequence of $P$ transformable patches $[\mathbf{o}_{l_p:r_p}]_{p=1}^P$. Each patch is defined by a consistent time window of size $s$ (e.g., 2 hours), ensuring a unified temporal resolution across all variables regardless of their sampling density. For the $p$-th patch, the boundary indices $l_p$ and $r_p$ are determined by the time horizon:

$$l_p = \min i \mid t_i \geq (p-1)s, \quad r_p = \max i \mid t_i < ps. \quad (8)$$

**Continuous Time Embedding.** To accurately represent the continuous-time nature of IMTS, we adopt a continuous time embedding $\phi(t)$ to encode each timestamp $t_l$. The embedding is formulated as:

$$\phi(t)[d] = \begin{cases} \omega_0 \cdot t + \alpha_0, & \text{if } d = 0 \\ sin(\omega_d \cdot t + \alpha_d), & \text{if } 0 < d < D_t \end{cases}, \quad (9)$$

where $\omega_d$ and $\alpha_d$ are learnable parameters. The linear term captures non-periodic trends, while the periodic terms model the inherent seasonality within the data. Each observation $i$ within a patch is represented as a concatenated vector:

$$\mathbf{z}_{l_p:r_p} = [z_i]_{i=l_p}^{r_p} = [\phi(t_i) \parallel v_i]_{i=l_p}^{r_p}. \quad (10)$$

**Transformable Time-aware Convolution.** Since each transformable patch is essentially a sub-irregular time series, we utilize the Transformable Time-aware Convolution Network (TTCN) (Zhang et al., 2024d) to capture its local semantics. TTCN is specifically designed to handle the inherent variability of IMTS by providing the flexibility to adapt to variable-length sequences and varying time intervals without additional learnable parameters. By utilizing transformable filters that ensure consistent scaling across diverse temporal resolutions, the latent patch embedding is extracted as:

$$h_p^c = \text{TTCN}(\mathbf{z}_{l_p:r_p}). \quad (11)$$

Considering that some patches may lack observations in sparse scenarios, we incorporate a patch masking term $m_p$ (where $m_p = 1$ if the patch contains observations, else 0) to produce the final encoding $h_p = [h_p^c \parallel m_p] \in \mathbb{R}^D$, and we have $h_{\text{patch}} = [h_p]_{p=1}^P \in \mathbb{R}^{P \times D}$.

### 4.3.2. QUERY-BASED PATCH MIXING

**Query-based Patch Attention.** Directly processing a sequence of $P$ patches is often inefficient in the context of IMTSF due to extreme variability in information density. Patches corresponding to sparse sampling intervals contain fewer observations, which may introduce significant noise and hinder the model's ability to capture meaningful temporal patterns. To distill the most salient features and ensure a fixed-capacity representation regardless of the input sequence length, we introduce a bottleneck mechanism via $W$ learnable query patches $\mathbf{W} \in \mathbb{R}^{W \times D}$.

These learnable queries interact with the extracted patch features $\mathbf{h}_{\text{patch}} = [h_1, \ldots, h_P] \in \mathbb{R}^{P \times D}$ through a cross-attention mechanism. This process allows the model to adaptively aggregate information from the most informative temporal patches while suppressing noise from sparse patches. The attention weights $\mathbf{A}$ are computed as:

$$\mathbf{A} = \text{Softmax}\left(\frac{\mathbf{W}(\mathbf{h}_{\text{patch}} + \mathbf{PE})^\top}{\sqrt{D}}\right) \in \mathbb{R}^{W \times P}, \quad (12)$$

where $\mathbf{PE} \in \mathbb{R}^{P \times D}$ represents the positional encodings assigned to each patch to preserve temporal order. The final semantic representation $\mathbf{h}_W \in \mathbb{R}^{W \times D}$ is obtained by taking the weighted sum of the patch features:

$$\mathbf{h}_W = \mathbf{A}\mathbf{h}_{\text{patch}}. \quad (13)$$

**Dual-Mixing Blocks.** The condensed semantic representations $\mathbf{H}_W \in \mathbb{R}^{N \times W \times D}$ (where $N$ is the number of variables) are further processed through $L$ layers of Dual-Mixing Blocks to capture deep temporal and multivariate dependencies. These blocks alternate between mixing information within the temporal domain of individual variables and across the variable dimension.
*Patch Mixing.* To capture the intra-variable temporal evolution, the patch mixer operates across the $W$ semantic patches. For each variable $n$ and feature dimension $d$, the MLP-based patch mixer allows different semantic tokens to interact:

$$\mathbf{H}' = \text{LN}(\mathbf{H}_W + \text{Permute}(\text{MLP}_{\text{patch}}(\text{Permute}(\mathbf{H}_W)))), \quad (14)$$

where $\text{LN}(\cdot)$ denotes Layer Normalization, $\text{MLP}_{\text{patch}}(\cdot)$ consists of two linear layers with a GELU non-linearity. By mixing information across the distilled $W$ tokens, the model captures the high-level temporal dynamics that characterize each specific variable.

*Variable Mixing.* To model inter-variable correlations and joint system dynamics, we employ a variable mixer that operates across the $N$ variables for each semantic token and feature dimension:

$$\mathbf{H}'' = \text{LN}(\mathbf{H}' + \text{Permute}(\text{MLP}_{\text{var}}(\text{Permute}(\mathbf{H}')))), \quad (15)$$

This operation allows the model to exchange information among different variables, capturing complex cross-channel dependencies which are crucial for accurate multivariate forecasting.

Following the $L$ dual-mixing layers, the local time module employs an aggregation layer to consolidate the $W$ semantic dimensions into a fixed-length temporal context vector. Depending on the configuration, this is implemented as either a linear projection or a 1D-convolutional layer:

$$\mathbf{h}_{\text{time}} = \text{Aggregation}(\text{Flatten}(\mathbf{H}'')) \in \mathbb{R}^{N \times D}, \quad (16)$$

where $\mathbf{H}'' \in \mathbb{R}^{N \times W \times D}$, $\mathbf{h}_{\text{time}}$ serves as the final local temporal representation, encapsulating the fine-grained patterns and inter-variable interactions extracted from the non-uniform input patches.

## 4.4. Output Module

The Output Module facilitates the joint fusion of time and frequency features, employing an MLP decoder to generate the initial predictions. The final predictions is synthesized by combining the initial predictions with the seasonal bias derived from future timestamps via Inverse NUDFT, followed by a Masked Denormalization step.

**Joint Feature Fusion.** Features from both branches are fused via a learnable weighted connection:

$$\mathbf{h}_{\text{joint}} = \mathbf{h}_{\text{time}} + \lambda_{\text{fusion}} \cdot \mathbf{h}_{\text{freq}}, \quad (17)$$

where $\lambda_{\text{fusion}}$ is a learnable scaling parameter.

**MLP Decoder and Seasonal Bias**. The base prediction $\hat{\mathcal{V}}_{\text{base}}$ is generated by passing $\mathbf{h}_{\text{joint}}$ and future time encodings $Q_{\text{Embedding}}$ through an MLP decoder.

$$\hat{\mathcal{V}}_{\text{base}} = \text{MLP}([\mathbf{h}_{\text{joint}} \| Q_{\text{Embedding}}]). \quad (18)$$

where $Q_{\text{Embedding}}$ denotes the encoded representation of the future timestamps $Q$, derived via the transformation defined in Equation 9.

Concurrently, Inverse NUDFT is performed using the spectral coefficients $\mathbf{S}_{\text{refined}}$ to estimate the seasonal bias:

$$\hat{\mathcal{V}}_{\text{bias}} = \sum_{k=1}^{K} [\hat{R}_n(\omega_k) \cos(2\pi\omega_k Q) - \hat{I}_n(\omega_k) \sin(2\pi\omega_k Q)]. \quad (19)$$

The final prediction is obtained as:

$$\hat{\mathcal{V}} = \text{Denorm}(\hat{\mathcal{V}}_{\text{base}} + \lambda_s \cdot \hat{\mathcal{V}}_{\text{bias}}), \quad (20)$$

where $\lambda_s$ is a learnable scaling parameter.

## 4.5. Loss Function

The optimization of TFMixer balances supervised forecasting with self-supervised reconstruction. The total loss $\mathcal{L}$ is formulated as:

$$\mathcal{L} = \mathcal{L}_{\text{fore}}(\hat{\mathcal{V}}, \mathcal{V}_{\text{true}}) + \gamma \mathcal{L}_{\text{recon}}(\mathcal{X}, \dot{\mathcal{X}}), \quad (21)$$

$$\dot{\mathcal{X}} = \sum_{k=1}^{K} [\hat{R}_n(\omega_k) \cos(2\pi\omega_k \mathcal{T}) - \hat{I}_n(\omega_k) \sin(2\pi\omega_k \mathcal{T})], \quad (22)$$

where $\gamma$ is a hyperparameter that weights the relative importance of the two objectives.

Forecasting Loss ($\mathcal{L}_{\text{fore}}$) evaluates the predictive performance on future target values using the Mean Absolute Error (MAE). To accommodate the irregular sampling and missing values inherent in IMTS, the loss is computed exclusively over the observed future timestamps. This objective restricts the model's focus to minimizing errors solely on valid target queries.

Reconstruction Loss ($\mathcal{L}_{\text{recon}}$) measures the MAE between the original historical observations $\mathcal{X}$ and the reconstructed historical values $\dot{\mathcal{X}}$. As defined, $\dot{\mathcal{X}}$ is synthesized from the refined spectral coefficients $\{R_n(\omega_k), I_n(\omega_k)\}$ via the Inverse NUDFT over the historical timestamps $\mathcal{T}$. This objective ensures that the Global Frequency Module captures authentic physical periodicities rather than spurious noise.

# 5. Experiments

In Section 5.1, we introduce the datasets, baselines, and implementation details. Section 5.2 presents the primary experimental results, while Section 5.3 provides detailed analyses, including ablation studies 5.3.1, parameter sensitivity analyses 5.3.2, and extended forecasting horizon experiments 5.3.3. Due to space limitations, the experiments on varying lookback lengths are provided in Appendix C.2.

## 5.1. Experimental Settings

### 5.1.1. DATASETS

To ensure comprehensive and fair comparisons across different models, we perform experiments on four widely used benchmarks for irregular multivariate forecasting: PhysioNet, MIMIC, HumanActivity, and USHCN. These datasets span multiple domains such as healthcare, biomechanics, and climate science. For PhysioNet, MIMIC-III, and USHCN, we follow the preprocessing setup in previous work (Yalavarthi et al., 2024; Li et al., 2025). For Human Activity, we follow the preprocessing set up in (Zhang et al., 2023; Li et al., 2025). All datasets are partitioned into training, validation, and test sets using a standard 80%, 10%, and 10% ratio, respectively. Summary statistics of the datasets

*Table 1.* Forecasting performance on four IMTS datasets. Overall performance is evaluated by MSE and MAE (mean ± std). The best and second-best results are highlighted in **bold** and with an underline, respectively.

| Dataset | HumanActivity | | USHCN | | PhysioNet | | MIMIC | |
|---|---|---|---|---|---|---|---|---|
| Metric | MSE | MAE | MSE | MAE | MSE | MAE | MSE | MAE |
| PrimeNet | 4.2507±0.0041 | 1.7018±0.0011 | 0.4930±0.0015 | 0.4954±0.0018 | 0.7953±0.0000 | 0.6859±0.0001 | 0.9073±0.0001 | 0.6614±0.0001 |
| NeuralFlows | 0.1722±0.0090 | 0.3150±0.0094 | 0.2087±0.0258 | 0.3157±0.0187 | 0.4056±0.0033 | 0.4466±0.0027 | 0.6085±0.0101 | 0.5306±0.0066 |
| CRU | 0.1387±0.0073 | 0.2607±0.0092 | 0.2168±0.0162 | 0.3180±0.0248 | 0.6179±0.0045 | 0.5778±0.0031 | 0.5895±0.0092 | 0.5151±0.0048 |
| mTAN | 0.0993±0.0026 | 0.2219±0.0047 | 0.5561±0.2020 | 0.5015±0.0968 | 0.3809±0.0043 | 0.4291±0.0035 | 0.9408±0.1126 | 0.6755±0.0459 |
| SeFT | 1.3786±0.0024 | 0.9762±0.0007 | 0.3345±0.0022 | 0.4083±0.0084 | 0.7721±0.0021 | 0.6760±0.0029 | 0.9230±0.0015 | 0.6628±0.0008 |
| GNeuralFlow | 0.3936±0.1585 | 0.4541±0.0841 | 0.2205±0.0421 | 0.3286±0.0412 | 0.8207±0.0310 | 0.6759±0.0100 | 0.8957±0.0209 | 0.6450±0.0072 |
| GRU-D | 0.1893±0.0627 | 0.3253±0.0485 | 0.2097±0.0493 | 0.3045±0.0305 | 0.3419±0.0029 | 0.3992±0.0011 | 0.4759±0.0100 | 0.4526±0.0055 |
| Raindrop | 0.0916±0.0072 | 0.2114±0.0072 | 0.2035±0.0336 | 0.3029±0.0264 | 0.3478±0.0019 | 0.4044±0.0020 | 0.6754±0.1829 | 0.5444±0.0868 |
| Warpformer | 0.0449±0.0010 | 0.1228±0.0018 | 0.1888±0.0598 | 0.2939±0.0591 | **0.3056±0.0011** | 0.3661±0.0016 | 0.4302±0.0035 | 0.4025±0.0014 |
| tPatchGNN | 0.0443±0.0009 | 0.1247±0.0031 | 0.1885±0.0403 | 0.3084±0.0479 | 0.3133±0.0053 | 0.3697±0.0049 | 0.4431±0.0115 | 0.4077±0.0088 |
| GraFITi | 0.0437±0.0005 | 0.1221±0.0017 | 0.1691±0.0093 | 0.2777±0.0248 | 0.3075±0.0015 | 0.3637±0.0036 | 0.4359±0.0455 | 0.4142±0.0297 |
| Hi-Patch | 0.0435±0.0002 | 0.1204±0.0009 | 0.1749±0.0268 | 0.2717±0.0216 | 0.3071±0.0029 | 0.3675±0.0042 | 0.4279±0.0010 | 0.4033±0.0032 |
| KAFNet | 0.0429±0.0003 | 0.1161±0.0010 | 0.1698±0.0181 | 0.2690±0.0226 | 0.3164±0.0028 | 0.3715±0.0038 | 0.4402±0.0086 | 0.4102±0.0041 |
| APN | 0.0421±0.0001 | 0.1159±0.0006 | 0.1590±0.0137 | 0.2611±0.0167 | 0.3093±0.0011 | 0.3650±0.0026 | 0.4292±0.0027 | 0.4016±0.0016 |
| **TFMixer** | **0.0415±0.0004** | **0.1124±0.0005** | **0.1448±0.0044** | **0.2052±0.0036** | 0.3118±0.0023 | **0.3576±0.0019** | **0.4182±0.0038** | **0.3782±0.0034** |

*Table 2.* Dataset statistics.

| Dataset | # Variables | # samples | Avg # Obs. | Max Length |
|---|---|---|---|---|
| PhysioNet | 36 | 11,981 | 308.6 | 47 |
| MIMIC | 96 | 21,250 | 144.6 | 96 |
| HumanActivity | 12 | 1,359 | 362.2 | 131 |
| USHCN | 5 | 1,114 | 313.5 | 337 |

are provided in Table 2.

**PhysioNet** (Silva et al., 2012) The PhysioNet dataset consists of 12,000 irregular multivariate time series representing unique patients. Each entry comprises 41 clinical signals recorded sporadically during the initial 48 hours of ICU hospitalization. For experimental purposes, the data is partitioned into observation and prediction windows.

**MIMIC-III** (Johnson et al., 2016) The MIMIC-III dataset (Johnson et al., 2016) comprises 96 clinical variables for 23,457 patients during their first 48 hours in the ICU.

**Human Activity** The Human Activity dataset consists of 3D positional sensor data from five subjects, with 12 variables measured at irregular intervals. The time series are divided into 5,400 samples of 4,000 ms each.

**USHCN** (Menne et al., 2015) While the full USHCN dataset spans over 150 years of climate data from numerous U.S. stations across 5 variables, our study follows standard preprocessing by focusing on 1,114 stations from 1996 to 2000. This results in 26,736 IMTS instances.

### 5.1.2. BASELINES

We comprehensively evaluate our model against 14 baselines: 1) IMTS Classification/Imputation Models, comprising PrimeNet (Chowdhury et al., 2023), SeFT (Horn et al., 2020), mTAN (Shukla & Marlin, 2021), GRU-D (Che et al., 2016), Raindrop (Zhang et al., 2022), and Warpformer (Zhang et al., 2023); and 2) IMTS Forecasting Models, which include NeuralFlows (Bilos et al., 2021), CRU (Schirmer et al., 2022), GNeuralFlow (Mercatali et al., 2024), tPatchGNN (Zhang et al., 2024b), GraFITi (Yalavarthi et al., 2024), Hi-Patch (Luo et al., 2025), KAFNet (Zhou et al., 2026), and APN (Liu et al., 2026b). Detailed descriptions of the methods are provided in the Appendix A.1.

### 5.1.3. IMPLEMENTATION DETAILS

To keep consistent with previous works, we adopt Mean Squared Error (mse) and Mean Absolute Error (mae) as evaluation metrics. The lookback time periods are 36 hours for MIMIC-III, and PhysioNet, 3000 milliseconds for Human Activity, and 3 years for USHCN. Human Activity uses 300 milliseconds as forecast length, and the rest datasets use the next 3 timestamps as forecast targets, following the settings in existing works (Liu et al., 2026b; Li et al., 2025; Bilos et al., 2021; Brouwer et al., 2019). Following the settings in TFB (Qiu et al., 2024) and TAB (Qiu et al., 2025a), we do not apply the "Drop Last" trick to ensure a fair comparison. All our experiments are conducted on a server equipped with an NVIDIA Tesla-A800 GPU

*Table 3.* Ablation studies for TFMixer in terms of the lowest MSE and MAE highlighted in bold.

| Variations | Activity | | USHCN | | PhysioNet | | MIMIC | |
|---|---|---|---|---|---|---|---|---|
| Metric | MSE | MAE | MSE | MAE | MSE | MAE | MSE | MAE |
| w/o Global Frequency Module | 0.0430±0.0018 | 0.1147±0.0036 | 0.1468±0.0075 | 0.2145±0.0093 | 0.3123±0.0028 | 0.3580±0.0024 | 0.4211±0.0073 | 0.3805±0.0054 |
| w/o Local Time Module | 0.0432±0.0004 | 0.1147±0.0005 | 0.1566±0.0083 | 0.2339±0.0035 | 0.3542±0.0022 | 0.4023±0.0016 | 0.5454±0.0021 | 0.4635±0.0011 |
| w/o Reconstruction Loss | 0.0445±0.0011 | 0.1137±0.0011 | 0.1464±0.0040 | 0.2069±0.0040 | 0.3126±0.0015 | 0.3585±0.0023 | 0.4190±0.0048 | 0.3790±0.0039 |
| w/o Spectrum Refinement | 0.0420±0.0007 | 0.1126±0.0004 | 0.1519±0.0095 | 0.2059±0.0062 | 0.3128±0.0025 | 0.3578±0.0022 | 0.4197±0.0044 | 0.3789±0.0034 |
| w/o Query-based Patch Mixing | 0.0432±0.0004 | 0.1129±0.0005 | 0.1518±0.0116 | 0.2109±0.0091 | 0.3188±0.0023 | 0.3595±0.0022 | 0.4478±0.0005 | 0.3915±0.0006 |
| **TFMixer (Ours)** | **0.0415±0.0004** | **0.1124±0.0005** | **0.1448±0.0044** | **0.2052±0.0036** | **0.3118±0.0023** | **0.3576±0.0019** | **0.4182±0.0038** | **0.3782±0.0034** |

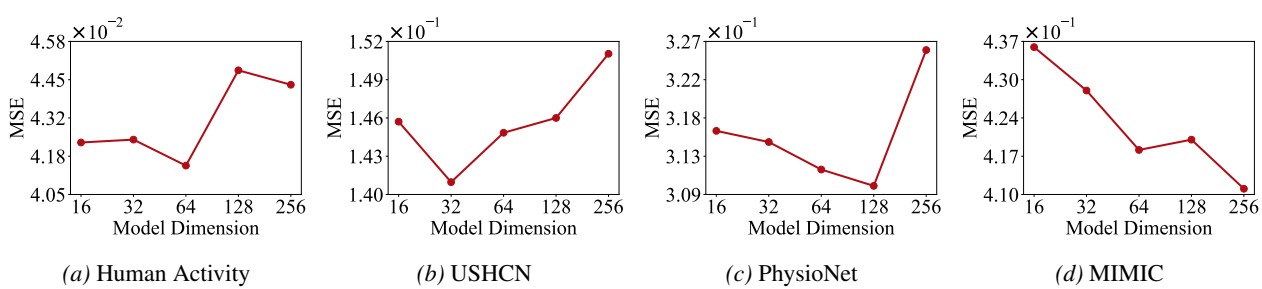

*(a)* Human Activity      *(b)* USHCN      *(c)* PhysioNet      *(d)* MIMIC

*Figure 4.* Parameter sensitivity studies of TFMixer, analyzing the impact of different model dimensions D.

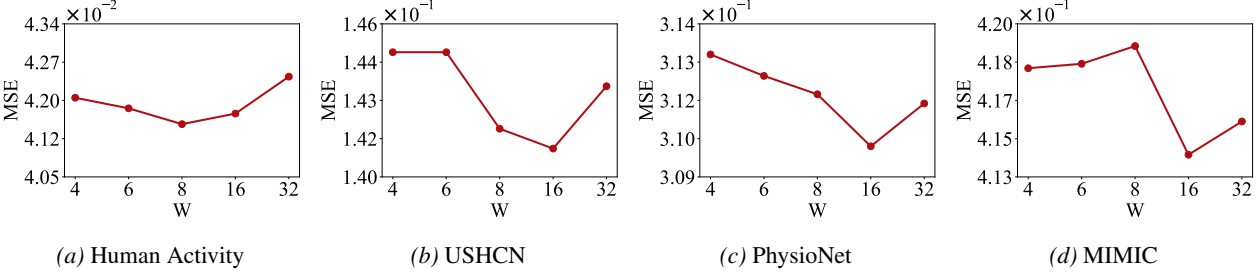

*(a)* Human Activity      *(b)* USHCN      *(c)* PhysioNet      *(d)* MIMIC

*Figure 5.* Parameter sensitivity studies of TFMixer, analyzing the impact of different number of learnable query patches W.

and implemented using the PyTorch 2.6.0+cu124 (Paszke et al., 2019) framework. The training process of TFMixer is guided by the L1 loss function and employs the ADAM optimizer. To ensure reproducibility and mitigate the effects of randomness, each experiment is run independently with five different random seeds (from 2024 to 2028), and we report the mean and standard deviation. Additionally, all datasets and code are available at: https://github.com/decisionintelligence/TFMixer.

### 5.2. Main Results

Comprehensive results are presented in Table 1 to demonstrate the performance of TFMixer. We have the following observations: 1) Compared with forecasters of various structures, TFMixer achieves outstanding predictive performance. In terms of absolute metrics, TFMixer surpasses the

second-best baseline, APN, with a 3.0% reduction in MSE and an 8.1% reduction in MAE. 2) TFMixer demonstrates exceptional cross-domain generalization and robustness, delivering superior performance across diverse IMTS datasets with distinct characteristics—ranging from healthcare (PhysioNet, MIMIC) to biomechanics (HumanActivity) and climate science (USHCN). This outstanding performance is primarily attributed to its decoupled architecture, which models global periodicity and local temporal representations in parallel. Specifically, TFMixer utilizes a learnable Non-Uniform Discrete Fourier Transform (NUDFT) to directly extract spectral representations from irregular timestamps, while employing a query-based patch attention mechanism to adaptively aggregate informative temporal segments and alleviate information density imbalance.

*Table 4.* Experimental results on four irregular multivariate time series datasets evaluated using MSE and MAE, with the lookback length following Table 1 and forecast horizons set to the rest length of the whole series, which are 12 hours for MIMIC-III and PhysioNet'12, 1000 milliseconds for Human Activity, and 1 year for USHCN.

| Dataset | HumanActivity | | USHCN | | PhysioNet | | MIMIC | |
|---|---|---|---|---|---|---|---|---|
| Metric | MSE | MAE | MSE | MAE | MSE | MAE | MSE | MAE |
| tPatchGNN | 0.0580±0.0011 | 0.1448±0.0027 | 0.5753±0.0892 | 0.4111±0.0713 | 0.3635±0.0014 | 0.4120±0.0020 | 0.5140±0.0040 | 0.4440±0.0052 |
| Hi-Patch | 0.0557±0.0002 | 0.1423±0.0011 | 0.4528±0.0075 | 0.3148±0.0099 | 0.3628±0.0018 | 0.4145±0.0028 | 0.5024±0.0115 | 0.4415±0.0035 |
| KAFNet | 0.0559±0.0002 | 0.1376±0.0012 | 0.5327±0.1160 | 0.3932±0.0731 | 0.3709±0.0018 | 0.4182±0.0015 | 0.5334±0.0253 | 0.4542±0.0134 |
| APN | 0.0657±0.0022 | 0.1577±0.0033 | 0.5455±0.1106 | 0.3894±0.0642 | 0.3686±0.0013 | 0.4159±0.0020 | 0.4982±0.0064 | 0.4421±0.0070 |
| **TFMixer** | **0.0553±0.0003** | **0.1350±0.0005** | **0.4444±0.0041** | **0.2614±0.0084** | **0.3603±0.0024** | **0.3978±0.0022** | **0.4981±0.0058** | **0.4095±0.0028** |

## 5.3. Model Analyses

### 5.3.1. ABLATION STUDIES

We perform ablation studies to validate the contribution of each primary component of TFMixer—see Table 3. We make the following observations: 1) Removing either the Global Frequency Module or the Local Time Module and retaining a single branch leads to performance degradation, demonstrating their complementarity and the necessity of jointly modeling global and local dependencies. 2) Eliminating the reconstruction loss (Section 4.5) and optimizing only with the forecasting loss degrades performance, indicating that the reconstruction loss acts as an effective regularizer for the Global Frequency Module. 3) Removing the Spectrum Refinement module (Section 4.2.1) and reverting to a purely linear NUDFT calculation impairs model accuracy. This suggests that the learnable NUDFT is critical for effectively extracting spectral representations from irregular time series. 4) Removing the Query-based Patch Mixing module (Section 4.3.2) and using only the naive Patch Encoder reduces forecasting accuracy, confirming that the query-based mechanism mitigates uneven information density across patches and improves predictive performance.

### 5.3.2. PARAMETER SENSITIVITY

We also conduct parameter sensitivity studies of TFMixer. We make the following observations: 1) Figure 4 shows the impact of the model dimension $D$ varies significantly with the dataset scale. Smaller datasets, such as Human Activity and USHCN, tend to perform better with lower dimensions (e.g., 32 or 64); excessive dimensionality in these cases often leads to performance degradation due to overfitting. Conversely, larger datasets like PhysioNet and MIMIC require higher dimensions (e.g., 128 or 256) to provide sufficient representational capacity for data fitting. 2) Figure 5 demonstrates the influence of the number of learnable query patches, $W$. We observed that both excessively small and large patch counts negatively impact performance. A small $W$ fails to capture sufficient semantic information, while an overly large $W$ results in fragmented and highly coupled semantic information. The optimal value typically resides at 8 or 16. Due to space constraints, please refer to the Appendix C.3 for additional parameter sensitivity analyses.

### 5.3.3. VARYING FORECASTING HORIZONS

We assess the robustness of TFMixer by evaluating its performance across varying forecast horizons, comparing it against competitive IMTS baselines.

For varying forecast horizons, we follow the experimental settings established in previous work (Zhang et al., 2024c) and maintain the lookback lengths consistent with Table 1. Specifically, the forecast horizons are set to 12 hours for MIMIC and PhysioNet'12, 1000 milliseconds for Human Activity, and 1 year for USHCN. The results are summarized in Table 4. It can be observed that TFMixer consistently achieves the best overall performance across most datasets. The stability of TFMixer in long-term forecasting is primarily attributed to its global frequency module, which provides a consistent periodic skeleton that remains robust even as the forecasting horizon extends and local temporal patterns become more uncertain.

## 6. Conclusion

In this paper, we present TFMixer, a novel joint time–frequency modeling framework specifically designed for IMTSF. By integrating a Global Frequency Module based on learnable NUDFT and a Local Time Module with a query-based patch mixing mechanism, TFMixer effectively overcomes the limitations of standard models in handling non-uniform sampling and information density imbalance. Our dual-branch approach enables the simultaneous capture of global dependencies and fine-grained local dynamics. Extensive evaluations on real-world datasets demonstrate that TFMixer consistently outperforms existing state-of-the-art methods. Additionally, all datasets and code are available at: https://github.com/decisionintelligence/TFMixer.

## Acknowledgements

This work was partially supported by National Natural Science Foundation of China (62472174), the ECNU Multifunctional Platform for Innovation (001), and the Fundamental Research Funds for the Central Universities (YBNLTS2026010). Jilin Hu is the corresponding author of the work.

## Impact Statement

This paper presents work whose goal is to advance the field of Machine Learning. There are many potential societal consequences of our work, none which we feel must be specifically highlighted here.

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

# A. Experiments setup details

## A.1. Baselines

**PrimeNet** (Chowdhury et al., 2023) is an IMTS pretraining model.

**NeuralFlows** (Bilos et al., 2021) This model uses continuous-time normalizing flows to model the density of trajectories, offering a more flexible alternative to Neural ODEs.

**CRU** (Schirmer et al., 2022) The Continuous Recurrent Unit models the hidden state transition as a continuous-time process, effectively bridging RNNs and SDEs.

**mTAN** (Shukla & Marlin, 2021) converts IMTS to reference points for IMTS classification.

**SeFT** (Horn et al., 2020) Set Functions for Time Series treat observations as an unordered set of points, utilizing attention to remain invariant to sampling frequency.

**GNeuralFlow** (Mercatali et al., 2024) enhances NeuralFlows with graph neural networks for IMTS analysis.

**GRU-D** (Che et al., 2016) A classic RNN variant that incorporates trainable decay terms to account for the time elapsed between irregular observations.

**Raindrop** (Zhang et al., 2022) A graph-based attention model that captures message-passing between sensors to handle missing values and irregular intervals.

**Warpformer** (Zhang et al., 2023) This model introduces a time-warping mechanism within a Transformer block to handle non-stationary and irregularly sampled sequences.

**tPatchGNN** (Zhang et al., 2024b) This framework partitions time series into temporal patches and uses graph neural networks to model inter-variable correlations.

**GraFITi** (Yalavarthi et al., 2024) uses bipartite graphs for IMTS forecasting.

**Hi-Patch** (Luo et al., 2025) A hierarchical patching approach that captures both local details and global trends by processing the series at multiple granularities.

**KAFNet** (Zhou et al., 2026) Uses a compact CPA-grounded architecture with frequency-domain attention to achieve state-of-the-art efficiency and accuracy.

**APN** (Liu et al., 2026b) Addresses irregularity via a Time-Aware Patch Aggregation (TAPA) module that adaptively regularizes sequences for efficient forecasting.

# B. Other Related Works

## B.1. Regular Time Series Forecasting Methods

Time series forecasting has become a fundamental task in numerous real-world applications, including economic analysis (Wu et al., 2024; Wang et al., 2025), intelligent transportation systems (Qiu et al., 2026; Wang et al., 2026b; Wu et al., 2026; Shao et al., 2025), healthcare analytics (Wu et al., 2025a; Wang et al., 2026c; Cheng et al., 2026), energy systems (Wang et al., 2026a; Liu et al., 2026a), and AIOps (Yu et al., 2025a). Given historical observations, accurately forecasting future values in advance is of great practical importance (Yu et al., 2025b; Qiu et al., 2025b;a; Wu et al., 2025c;b;d; Shang et al., 2026; 2024; Chen et al., 2023; Shang & Chen, 2024).

Early forecasting studies mainly relied on statistical approaches, such as ARIMA (Box & Pierce, 1970), ETS (Hyndman et al., 2008), and Theta (Garza et al., 2022). These methods have achieved considerable success in many traditional forecasting tasks due to their solid theoretical foundations and interpretability. Nevertheless, they generally depend on handcrafted feature engineering and strong prior assumptions about temporal distributions, which restrict their ability to capture highly nonlinear and complex temporal patterns encountered in modern large-scale applications (Fang et al., 2025; Li et al., 2026a; Zhang et al., 2024a; 2025b; Ni et al., 2026; Chen et al., 2025; Qi et al., 2025; Ma et al., 2024; 2025b; 2022; Lu et al., 2024b).

Recent advances in deep learning have significantly reshaped the landscape of time series forecasting (Fang et al., 2023; 2022; Li et al., 2026b; Zhang et al., 2025a;c; Wang et al., 2026d; Ma et al., 2026; 2025a; Yu et al., 2025d;c;e; Lu et al., 2024a; 2023). Deep neural networks are capable of automatically extracting informative temporal representations from raw sequential observations, enabling more flexible modeling of complicated temporal dependencies. As a result, a variety of neural forecasting architectures have been proposed in recent years (Yang et al., 2026; 2025b; Hu et al., 2026; Yang et al., 2025a). Among them, Transformer-based models, including Informer (Zhou et al., 2021), FEDformer (Zhou et al., 2022b), DUET (Qiu et al., 2025c), Autoformer (Wu et al., 2021), Triformer (Cirstea et al., 2022), and PatchTST (Nie et al., 2023), have shown strong capability in modeling long-term dependencies and complex token-level interactions. In parallel, lightweight MLP-based methods, such as SparseTSF (Lin et al., 2024b), CycleNet (Lin et al., 2024a), NLinear (Zeng et al., 2023), and DLinear (Zeng et al., 2023), reveal that competitive forecasting performance can also be achieved with substantially simpler architectures and fewer parameters.

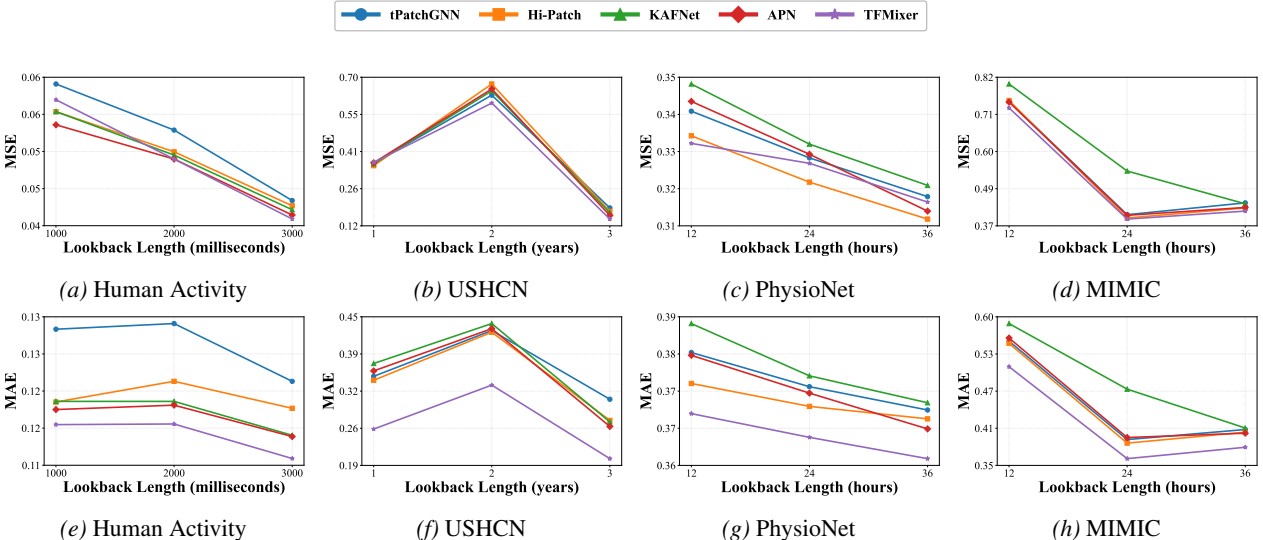

*Figure 6.* Forecasting performance with varying lookback lengths and fixed forecast horizons.

## C. Experiments

### C.1. Masked RevIN Module

To mitigate the adverse effects of distribution shift and non-stationarity in time series data, we employ the Masked Reversible Instance Normalization (Masked RevIN) module. Unlike standard normalization techniques that operate across the entire input, Masked RevIN is specifically designed to be compatible with irregular time series forecasting task. The module operates in two distinct phases:

**Masked Normalization**: Prior to entering the encoder, the module calculates the mean and standard deviation of each input instance. Crucially, to prevent the masked portions (typically represented by zero paddings or mask tokens) from biasing the statistical estimates, the module computes these metrics exclusively based on the visible (unmasked) time points. This ensures that the normalized representation faithfully reflects the underlying distribution of the actual data, even under high masking ratios.

**Masked Denormalization**: By utilizing the statistics previously stored during the normalization phase, it restores the output to its original magnitude and scale.

### C.2. Varying Lookback Lengths

For varying lookback lengths, the experimental results are illustrated in Figure 6. We maintain the forecast horizons used in 1 and vary the lookback lengths across three scales for each dataset: (1) 12, 24, and 36 hours for MIMIC and PhysioNet; (2) 1000, 2000, and 3000 milliseconds for Human Activity; and (3) 1, 2, and 3 years for USHCN. As shown in the figures, the forecasting performance of TFMixer generally

improves as the lookback length increases, benefiting from the richer historical context. Notably, TFMixer exhibits a more stable improvement trend compared to baselines like tPatchGNN, as its query-based patch attention can adaptively aggregate information from longer histories without being overwhelmed by the increased sparsity or noise. One exception occurs in the MIMIC dataset, where performance gains plateau or slightly diminish when the lookback length reaches 36 hours. This phenomenon suggests that for intensive care scenarios, excessively long historical windows might introduce distribution shifts or irrelevant temporal dependencies that challenge current modeling paradigms.

### C.3. Parameter Sensitivity

We also conduct parameter sensitivity studies of TFMixer. We make the following observations: 1) Figure 7 shows the impact of the model dimension $D$ varies significantly with the dataset scale. Smaller datasets, such as Human Activity and USHCN, tend to perform better with lower dimensions (e.g., 32 or 64); excessive dimensionality in these cases often leads to performance degradation due to overfitting. Conversely, larger datasets like PhysioNet and MIMIC require higher dimensions (e.g., 128 or 256) to provide sufficient representational capacity for data fitting. 2) Figure 8 demonstrates the influence of the number of learnable query patches, $W$. We observed that both excessively small and large patch counts negatively impact performance. A small $W$ fails to capture sufficient semantic information, while an overly large $W$ results in fragmented and highly coupled semantic information. The optimal value typically resides at 8 or 16. 3) Figure 9 illustrates the sensitivity of TFMixer to the patch size $P$. We find that the model's performance

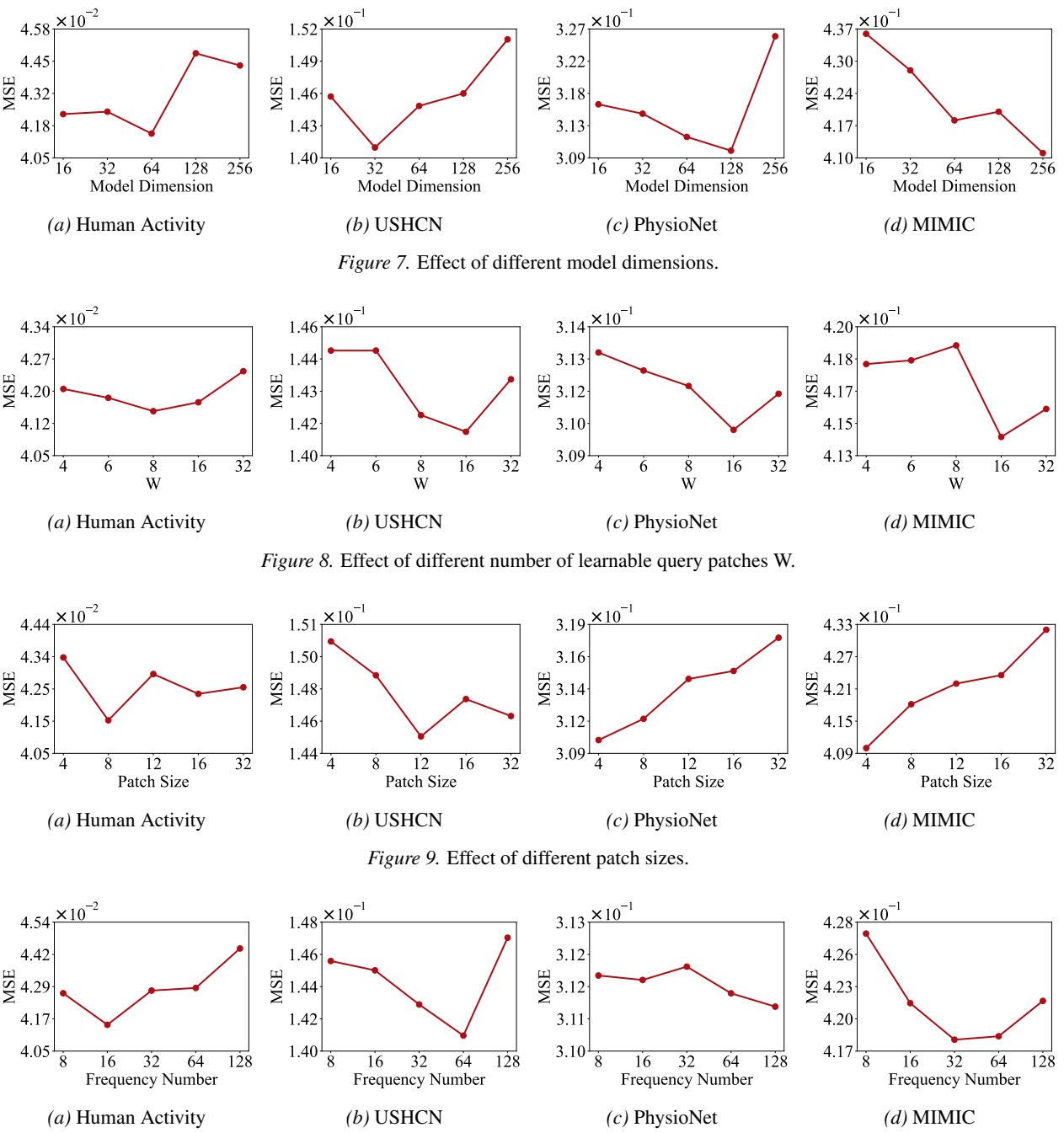

*Figure 7.* Effect of different model dimensions.

*Figure 8.* Effect of different number of learnable query patches W.

*Figure 9.* Effect of different patch sizes.

*Figure 10.* Effect of different frequency numbers.

is sensitive to the granularity of local temporal partitioning. When $P$ is too small (e.g., 4 or 8), the model tends to capture excessive local noise and increases the computational burden of the attention mechanism, potentially leading to sub-optimal results. On the other hand, a disproportionately large $P$ (e.g., 64) may cause the loss of fine-grained temporal dynamics within each patch, as it smooths out critical local fluctuations. The results indicate that a moderate patch size, such as 16 or 24, strikes a favorable balance be-

tween local detail preservation and computational efficiency across most datasets. 4) Figure 10 presents the impact of the number of selected frequencies $K$ in the global frequency module. $K$ determines the complexity of the global periodic structures the model can reconstruct. For datasets with prominent and complex periodicity, such as PhysioNet, a larger $K$ (e.g., 32 or 64) is generally beneficial for capturing high-frequency components that represent rapid physiological changes. However, for datasets with simpler underlying

patterns or higher noise levels, increasing $K$ beyond a certain threshold (e.g., 16) does not yield further gains and may even introduce spectral noise, leading to slight performance fluctuations. This suggests that TFMixer effectively concentrates the most significant energy in a small subset of the frequency spectrum, demonstrating the robustness of our NUDFT-based spectral extraction.

