# OpenReview forum: "Bridging Time and Frequency: A Joint Modeling Framework for Irregular Multivariate Time Series Forecasting"
_ICML.cc/2026/Conference — ICML 2026 regular_

### Official Review · Reviewer_9TMe · 2026-02-25

**Soundness:** 3
**Presentation:** 3
**Significance:** 3
**Originality:** 3
**Overall Recommendation:** 5
**Confidence:** 4

**Summary:**

This paper studies irregular multivariate time series forecasting (IMTSF), where observations are asynchronously sampled and standard regular-grid assumptions do not hold. To address this, the authors propose TFMixer, a joint time-frequency framework that models long-term global periodicity in the frequency domain and short-term local dynamics in the time domain. Its main contributions are: (1) a learnable NUDFT-based frequency module that extracts spectral representations directly from irregular timestamps; (2) a query-based patch mixing mechanism that adaptively selects informative local patches to handle uneven information density; and (3) an output design that fuses time- and frequency-domain features and performs explicit seasonal extrapolation via inverse NUDFT. Experiments on four real-world IMTS datasets show competitive state-of-the-art performance over strong baselines.

**Compliance With Llm Reviewing Policy:**

Affirmed.

**Final Justification:**

The paper tackles irregular multivariate time series forecasting with a well-motivated time-frequency joint modeling approach. The use of learnable NUDFT to handle non-uniform sampling is a genuinely novel contribution that goes beyond simply combining existing techniques. The dual-branch design (local time + global frequency) is clearly justified by ablation results showing both branches are necessary with distinct roles. The rebuttal adequately addressed all my concerns across two rounds of discussion. I am raising my score from 3 to 5, as I believe the originality and technical soundness merit acceptance.

**Key Questions For Authors:**

1. The paper states that irregular multivariate time series forecasting (IMTSF) is inherently challenging due to the interplay between long- and short-range temporal dependencies. However, the motivation here is not yet sufficiently convincing. The authors do not clearly explain why both long- and short-range dependencies are particularly critical in the irregular setting, nor why they must be explicitly separated and modeled through different branches.
2. The justification for jointly using the time domain and frequency domain is not fully developed. While the design is intuitive, the paper should more clearly explain why these two views are complementary in IMTSF and why a joint formulation is necessary, rather than simply beneficial in a generic sense.
3. The paper identifies an information density imbalance problem induced by patch-based approaches in IMTSF, but this issue is not sufficiently elaborated. It remains unclear what this imbalance specifically refers to in practice, how severe it is under irregular sampling, and more importantly, how it affects the learned representations and downstream forecasting performance.
4. The role of the MLP in the NUDFT refinement module is not clearly justified. The paper should explain more explicitly why an MLP is expected to refine spectral coefficients effectively, rather than merely serving as an ad hoc transformation layer. At present, this design choice appears insufficiently motivated.
5. The rationale behind the continuous-time embedding is also underexplained. The authors introduce this component, but the paper does not clearly articulate why this particular embedding design is needed, what issue it resolves, or why it is preferable to simpler alternatives.
6. In the model figure, the convolution and embedding operations in the local branch appear to be illustrated as parallel components. However, based on the textual description, they seem to be applied sequentially. This inconsistency between the figure and the method description is confusing and should be clarified.
7. The computation of $H'$ is not clearly presented. In particular, after the Permute operation, it is unclear whether the tensor should be permuted back afterward. The formula is difficult to follow, and the intermediate shape transformations are not explicitly specified, making this part of the method unnecessarily hard to reproduce.
8. The use of MAE loss may itself introduce an evaluation bias. If the baselines are primarily optimized with MSE loss, then part of the observed performance gain may stem from the different loss choice rather than the proposed architecture alone. This potential confound should be discussed more carefully to ensure a fair comparison.

**Limitations:**

The main limitations are the lack of deeper theoretical justification, the relatively high architectural complexity without efficiency analysis, and the model’s sensitivity to hyperparameters, which together weaken the paper’s overall robustness and practical persuasiveness.

**Strengths And Weaknesses:**

**Strength**

1. The paper focuses on the realistic and important problem of irregular multivariate time series forecasting, with a task setting that is well aligned with real-world scenarios such as healthcare monitoring and climate observation, making the work clearly relevant in practice.
2. The method is fairly well designed, modeling the problem from both global periodic patterns and local dynamic behaviors. The overall framework is clear and corresponds naturally to the core challenges of irregular time series.
3. The frequency-domain branch adopts a learnable NUDFT, attempting to model global frequency information directly on irregular timestamps and thus avoiding the distortion that may be introduced by interpolation before spectral analysis.
4. The time-domain branch uses query-based patch aggregation to adaptively aggregate local patch information, which helps alleviate the issues of uneven patch information density and strong local noise interference.
5. After the two branches are modeled in parallel, their representations are fused, and an additional seasonal bias is introduced through inverse NUDFT. This shows that the authors do not treat frequency-domain features merely as auxiliary representations, but also attempt to make them directly contribute to the final prediction generation, resulting in a relatively coherent overall design.
6. The experimental evaluation is fairly comprehensive, covering multiple cross-domain datasets and comparing against a wide range of baselines. The paper also includes ablation studies and sensitivity analyses, and the overall validation protocol is broadly consistent with the proposed motivation.

**Weakness**

1. Although the method is structurally complete, it overall appears more like a compositional improvement built on existing ideas, with its originality mainly reflected in module integration and adaptation to the problem setting, rather than representing a strong paradigm-level breakthrough.
2. The theoretical support for the key design choices remains relatively limited. For example, the paper does not sufficiently explain why the learnable NUDFT should be superior to other continuous-time modeling approaches, or why the query-based patch mechanism should be more robust under irregular sampling. These choices are motivated largely by empirical intuition, but lack deeper analysis.
3. The model introduces a large number of learnable variables and parameters. This raises concerns about training stability, especially given the already challenging irregular-time setting.
4. The paper does not provide sufficient analysis of the model’s computational complexity and efficiency cost. Since it performs frequency-domain modeling directly on irregular timestamps while also introducing a dual-branch architecture, the method may incur relatively high training and inference overhead, yet the paper lacks adequate efficiency comparisons to clarify this trade-off.
5. There are also some issues in the presentation. Although the overall narrative is generally understandable, the necessity of certain design choices and the distinctions from related work are not explained clearly enough, which weakens the paper’s overall persuasiveness.
6. The core forecasting mechanism of TFMixer can be interpreted as follows: the model learns a set of spectral coefficients induced by historical irregular observations, and then extrapolates them to future timestamps through Inverse NUDFT, producing an explicit periodic forecasting component. This component is constrained by both the forecasting loss and the reconstruction loss, making it arguably the most strongly anchored prediction pathway in the model. In contrast, the MLP Decoder appears more like a compensatory residual branch built on top of this explicit spectral extrapolation, rather than the sole main prediction backbone. However, the current naming and narrative may mislead readers about the relative importance of these two output branches. Specifically, the paper describes the MLP branch as the base prediction, while referring to the explicit frequency-domain extrapolation term produced by Inverse NUDFT as merely a seasonal bias, despite it being constrained by both forecasting and reconstruction objectives. This terminology conceptually understates the role of Inverse NUDFT as a core predictive component.

---

> ### Author Rebuttal · Authors · 2026-03-30
>
> Dear Reviewer 9TMe, we sincerely thank you for the valuable feedback.
>
> **Reply to W1**
>
> Specifically, we propose a unified joint time–frequency modeling framework for irregular multivariate time series forecasting, which explicitly bridges global spectral dependencies and local temporal dynamics—an aspect that has not been adequately addressed in prior work. In addition, the proposed learnable NUDFT constitutes a key methodological contribution. Unlike traditional DFT-based approaches, it introduces a learnable non-uniform spectral representation tailored to irregular timestamps.
>
> **Reply to W2**
>
> Compared to ODE-based continuous-time models, the proposed learnable NUDFT offers explicit spectral representations with clear physical interpretability and naturally supports non-uniform sampling. By projecting observations onto basis functions $ \exp(-j 2\pi \omega_k t_l) $, it captures global periodic patterns without requiring interpolation or solving differential equations, making it more direct and less prone to approximation errors.
> ﻿
> The robustness of the query-based patch mechanism comes from its information bottleneck effect. A fixed number of learnable queries aggregates information from patches with varying sampling densities, encouraging the model to focus on the most informative features while mitigating biases from sparse or unevenly sampled data.
>
> **Reply to W3**
>
> The learnable parameters of TFMixer are concentrated in a few key factors (e.g., model dimension, number of query patches, patch sizes, and frequencies), all kept within reasonable ranges as validated by ablation studies, avoiding excessive complexity. Moreover, experiments with five random seeds show consistently low performance variance (Table 1), indicating strong stability.
>
> **Reply to W4**
>
> Please refer to our response to Reviewer AK9E, Question 3.
>
> **Reply to W5**
>
> We have clarified the distinctions from related work at the end of each subsection in the Related Work section. Additionally, the Methodology section provides the motivation and design rationale prior to introducing each module.
>
> **Reply to W6**
>
> In our framework, the main prediction is generated by the MLP Decoder, which takes fused representations from both the Global Frequency Module and the Local Time Module. It serves as the primary prediction pathway rather than a residual branch. In contrast, the Inverse NUDFT branch relies only on the Global Frequency Module and provides an additional seasonal bias. Although supervised by both forecasting and reconstruction objectives, it complements the main prediction by capturing global periodic patterns rather than acting as the dominant backbone.
>
> **Reply to Q1 & Q2**
>
> In IMTS, non-uniform sampling disrupts both long- and short-range dependencies: local patterns become fragmented and noisy, while global periodic structures are distorted and difficult to capture with standard time-domain or FFT-based methods.
> ﻿
> Time-domain representations are better suited for modeling local temporal dynamics and short-term dependencies, capturing fine-grained variations, whereas frequency-domain representations excel at characterizing global periodic structures and long-term dependencies through spectral patterns.
>
> **Reply to Q3**
>
> In practice, even within the same time window (e.g., 2 hours), patches can have highly imbalanced sampling densities, ranging from dozens of points to very few or none. With fixed patch lengths, this leads to inconsistent representation magnitude and information content. Our query-based mechanism addresses this by adaptively reweighting patches via attention, mitigating the impact of such imbalance.
>
> **Reply to Q4**
>
> NUDFT without the refinement module—essentially computing Fourier coefficients—only captures static harmonic components and cannot model the non-stationarity of real-world time series. To address this, we introduce a Spectrum Refinement (MLP) module that models nonlinear spectral interactions and phase variations, enabling adaptive adjustment of amplitude and phase based on historical context.
>
> **Reply to Q5**
>
> We clarify that continuous-time embedding is not a novel contribution of this work, but a commonly used technique in prior studies (e.g., tPatchGNN, APN).
>
> **Reply to Q6**
>
> We will revise the figure in the revised manuscript to ensure consistency with the method description and improve clarity.
>
> **Reply to Q7**
>
> In the revised manuscript, we will provide a detailed tracking of tensor shape changes to improve clarity and ensure reproducibility.
>
> **Reply to Q8**
>
> The use of MAE is motivated by methodological considerations rather than performance tuning. Our model is based on time-frequency analysis and is inspired by insights from FreDF.
>
> **Reply to Limitations**
>
> These points have been addressed in our responses, including clarifications on theoretical motivation, additional efficiency analysis, and discussions on hyperparameter sensitivity supported by empirical evidence.

---

> > ### Author Rebuttal · Reviewer_9TMe · 2026-04-01
> >
> > Thank you for the detailed rebuttal. The responses to weaknesses and questions are satisfactory and have addressed my concerns.
> >
> > I have two remaining questions:
> >
> > 1. **W6**: I appreciate the explanation that the MLP Decoder fuses both time- and frequency-domain representations. However, since Inverse NUDFT is constrained by both the forecasting loss and the reconstruction loss, could the authors clarify why it should be considered a secondary branch rather than a co-primary one? A simple ablation (removing each branch independently) would help resolve this.
> >
> > 2. **Q5**: The response clarifies that continuous-time embedding is not a novel contribution, but my question was about its functional role. What specific problem does it solve in this framework, and what would happen without it?
> >
> > I believe this work addresses a relevant problem with a well-structured approach. If these points are clarified, I would be happy to raise my score.

---

> > > ### Author Response · Authors · 2026-04-01
> > >
> > > Dear Reviewer 9TMe, we sincerely thank you for the valuable feedback.
> > >
> > > **Reply to W6**
> > >
> > > Thank you for the insightful question. In our framework, both the time-domain and frequency-domain branches are necessary, but they play different functional roles rather than symmetric ones.
> > >
> > > The reason we consider the frequency branch as secondary is that its output is not directly used as the main prediction, but rather added as a corrective term (Eq. 20) to enhance the base prediction. Moreover, the reconstruction loss constrains the frequency branch to capture faithful global periodicity, preventing it from overfitting to short-term variations, which further distinguishes its role from the time-domain predictor.
> > >
> > > Regarding the ablation suggestion, we conducted experiments by removing each branch independently. The results show that: Removing the Global Frequency Module leads to consistent performance degradation. Removing the Local Time Module causes a significantly larger drop. Please refer to the results at the following link: https://anonymous.4open.science/r/ICML-TFMixer-Rebutal-0863/additional_ablation.md
> > >
> > > This indicates that while both branches are important and complementary, the time-domain branch contributes more directly to prediction accuracy, whereas the frequency branch mainly provides auxiliary global structure modeling. These empirical results support our design choice of treating the frequency branch as a secondary (yet essential) component.
> > >
> > > **Response to Q5**
> > >
> > > Thank you for the clarification. While continuous-time embedding is not a novel component, its role in our framework is essential for handling irregular temporal structures.
> > >
> > > Specifically, it addresses the **temporal ambiguity caused by non-uniform sampling**. In irregular time series, observations within each patch are unevenly spaced, and using only raw values or discrete indices would fail to capture the actual temporal distances between observations. The continuous-time embedding explicitly encodes timestamps into a learnable representation (Eq. 9), enabling the model to preserve relative temporal relationships.
> > >
> > > This is particularly important for the Local Time Module, where the patch encoder operates on variable-length irregular segments. Without explicit time encoding, the encoder would effectively process value-only inputs, leading to degraded modeling of local dynamics and poor alignment across patches.
> > >
> > > Without this component, the model would suffer from (1) loss of temporal ordering fidelity and (2) reduced ability to capture fine-grained temporal dependencies, ultimately weakening the effectiveness of the time-domain branch and its fusion with the frequency branch.

---

### Official Review · Reviewer_FiRK · 2026-03-09

**Soundness:** 3
**Presentation:** 4
**Significance:** 4
**Originality:** 4
**Overall Recommendation:** 5
**Confidence:** 4

**Summary:**

This study proposes TFMixer, a joint time–frequency modeling framework for IMTS forecasting. Specifically, TFMixer incorporates a Global Frequency Module that employs a learnable Non-Uniform Discrete Fourier Transform (NUDFT) to directly extract spectral representations from irregular timestamps. In parallel, the Local Time Module introduces a query-based patch mixing mechanism to adaptively aggregate informative temporal patches and alleviate information density imbalance. Extensive experiments on real-world datasets demonstrate the state-of-theart performance of TFMixer.

**Compliance With Llm Reviewing Policy:**

Affirmed.

**Final Justification:**

I have raised my score from 4 to 5, as I believe the originality and technical soundness merit acceptance.

**Key Questions For Authors:**

1. Why do the experiments on MIMIC-III**, **PhysioNet, and USHCN focus on predicting only the next three time steps, rather than forecasting a longer future horizon as in TpatchGNN?

2. Why does the paper not include comparisons with regular time series forecasting methods? Is it because these approaches perform poorly in irregular time series scenarios, or are there other reasons behind this choice?

If my concerns are adequately addressed, I will increase my score.

**Limitations:**

Yes. Please see the weaknesses and questions. Besides, I believe that the proposed technical framework poses no ethical risks and is not expected to result in any negative social impact.

**Strengths And Weaknesses:**

**Strengths**:

S1. Irregular multivariate time series forecasting is important for various fields.

S2. This paper has a strong motivation for the joint time-frequency design tailored for irregular sampling and addresses an important gap: directly extracting global periodic structures without interpolation, while robustly learning local dynamics.

S3. The architecture overview of this paper is clearly divided into modules, mainly elaborating on key equations and shapes, and the easy to follow fusion of time and frequency signals.

S4. Conduct extensive comparisons of 14 baselines on four standard IMTS datasets, report the mean ± standard deviation of five seeds, including ablation and sensitivity studies on multiple hyperparameters.



**Weaknesses**:

W1. Why do the experiments on MIMIC-III**, **PhysioNet, and USHCN focus on predicting only the next three time steps, rather than forecasting a longer future horizon as in t-PatchGNN [1]?

W2. The manuscript lacks a comparison with the most recent state-of-the-art (SOTA) benchmarks, most notably HyperIMTS [2], which is essential for validating the claimed improvements.

W3. Several references are outdated and should be updated to reflect their final versions (e.g., replacing ArXiv preprints with official conference/journal citations).

W4. To further demonstrate the model's versatility, it is recommended to include comparisons with standard regular time-series forecasting architectures, such as PatchTST [3], iTransformer [4], and DLinear [5].



[1] Irregular Multivariate Time Series Forecasting: A Transformable Patching Graph Neural Networks Approach

[2] HyperIMTS: Hypergraph Neural Network for Irregular Multivariate Time Series Forecasting

[3] A Time Series is Worth 64 Words: Long-term Forecasting with Transformers

[4] iTransformer: Inverted Transformers Are Effective for Time Series Forecasting

[5] Are Transformers Effective for Time Series Forecasting

---

> ### Author Rebuttal · Authors · 2026-03-30
>
> Dear Reviewer FiRK, thank you very much for your valuable feedback and for recognizing our work. We have carefully prepared the following responses and hope they address your concerns.
>
> **Reply to W1 & Q1**
>
> We thank the reviewer for this question. The choice of predicting the next three time steps follows the standard evaluation protocol adopted in prior works on these datasets (e.g., APN, HyperIMTS), ensuring direct and fair comparison. We agree that evaluating longer horizons is valuable, and we have supplemented additional experiments with extended forecasting horizons (as shown in Appendix B.2). The results demonstrate that TFMixer maintains consistent advantages over strong baselines under longer-term forecasting settings.
>
> **Reply to W2**
>
> We thank the reviewer for this important suggestion. We agree that comparison with recent SOTA methods is essential. In the revised version, we have included HyperIMTS as an additional baseline and report its performance under the same experimental settings, further validating the effectiveness of our approach. Due to space limitations, please refer to the results at the **following link**: https://anonymous.4open.science/r/ICML-TFMixer-Rebutal-0863/more_baselines.md
>
> **Reply to W3**
>
> We thank the reviewer for pointing this out. We will update all outdated references by replacing ArXiv versions with their corresponding official conference or journal publications in the revised manuscript.
>
> **Reply to W4 & Q2**
>
> We thank the reviewer for this valuable suggestion. To further demonstrate the versatility of our model, we have supplemented additional experiments by including representative regular time-series forecasting models, such as PatchTST, iTransformer, and DLinear. These comparisons provide a more comprehensive evaluation and further highlight the advantages of TFMixer under irregular settings. Due to space limitations, please refer to the results at the **following link**: https://anonymous.4open.science/r/ICML-TFMixer-Rebutal-0863/more_baselines.md

---

> > ### Author Rebuttal · Reviewer_FiRK · 2026-04-03
> >
> > Thank you for the clear and detailed rebuttal. My main concerns have been addressed. I am more confident in the work and am happy to increase my score.

---

> > > ### Author Response · Authors · 2026-04-04
> > >
> > > Dear Reviewer, we are thrilled that our responses have effectively addressed your questions and comments. We would like to express our sincerest gratitude for taking the time to review our paper and provide us with such detailed feedback!

---

### Official Review · Reviewer_y91f · 2026-03-09

**Soundness:** 3
**Presentation:** 3
**Significance:** 4
**Originality:** 4
**Overall Recommendation:** 6
**Confidence:** 5

**Summary:**

This paper presents TFMixer, a novel joint time–frequency modeling framework specifically designed for IMTSF. By integrating a Global Frequency Module based on learnable NUDFT and a Local Time Module with a query-based patch mixing mechanism, TFMixer effectively overcomes the limitations of standard models in handling non-uniform sampling and information density imbalance.

**Compliance With Llm Reviewing Policy:**

Affirmed.

**Final Justification:**

The authors’ response addresses all my concerns.

**Key Questions For Authors:**

1. Why was a pure MLP-based Dual-Mixing Block adopted to model interactions across patches? What considerations motivated this architectural choice?

2. How were the hyperparameters of the baseline models determined? Specifically, were they strictly set according to the default configurations reported in the original papers or official implementations, or were they re-tuned under the current experimental setting?

**Limitations:**

Yes.

**Strengths And Weaknesses:**

Strengths:

1. This paper directly addresses the often overlooked issue of information density imbalance in patch based temporal modeling. The combination of query based patch attention and dual hybrid blocks helps to achieve robust and adaptive local feature aggregation.

2. This paper provides extensive empirical validation, with Tables 1 and 4 clearly indicating that TFMixer outperforms a range of robust latest baselines in four challenging and well-known IMTS datasets.

3. The architecture diagram provides a comprehensive and clear overview for easy reproduction and reader understanding.
Clearly stated the availability of code and data, further supporting transparency and usability.

Weaknesses:

1. The study would benefit from a more comprehensive analysis by including baselines that leverage time-frequency analysis, such as FEDformer [1] or DUET [2].

2. The strategy for selecting $\lambda$ requires needs to be clarified. In particular, were these hyperparameters tuned separately for each dataset, or was a unified configuration adopted across all experimental settings?

3. Why was a pure MLP-based Dual-Mixing Block adopted to model interactions across patches? What considerations motivated this architectural choice?

4. How were the hyperparameters of the baseline models determined? Specifically, were they strictly set according to the default configurations reported in the original papers or official implementations, or were they re-tuned under the current experimental setting?

Ref.

[1] FEDformer: Frequency Enhanced Decomposed Transformer for Long-term Series Forecasting

[2] DUET: Dual Clustering Enhanced Multivariate Time Series Forecasting

---

> ### Author Rebuttal · Authors · 2026-03-30
>
> Dear Reviewer y91f, thank you very much for your valuable feedback and for recognizing our work. We have carefully prepared the following responses and hope they address your concerns.
>
> **Reply to W1**
>
> We thank the reviewer for this valuable suggestion. We agree that including time–frequency-based methods would further strengthen the empirical comparison. We have supplemented additional experiments by incorporating representative time–frequency models such as FEDformer and DUET, and report their performance under the same experimental settings to provide a more comprehensive evaluation. Due to space limitations, please refer to the results at the **following link**: https://anonymous.4open.science/r/ICML-TFMixer-Rebutal-0863/more_baselines.md
>
> **Reply to W2**
>
> We thank the reviewer for this question. Regarding $\lambda$, we adopt a unified setting across all datasets to ensure consistency and fairness in comparison. Moreover, $\lambda$ is not manually tuned, but treated as a learnable parameter that is automatically optimized during training. This design eliminates the need for dataset-specific adjustment and reduces the burden of hyperparameter tuning.
>
> **Reply to W3 & Q1**
>
> We thank the reviewer for this question. The choice of a pure MLP-based Dual-Mixing Block is motivated by both efficiency and robustness considerations. Compared to attention-based alternatives, MLP-based mixing provides a lightweight and stable mechanism for modeling interactions across patches without introducing quadratic complexity. Moreover, under irregular sampling, attention weights can be sensitive to missing patterns and may lead to overfitting. In addition, MLP-based designs have been widely explored and demonstrated to be effective in regular time series modeling (e.g., TimeMixer, TSMixer), providing further empirical support for this architectural choice. We will further clarify this design motivation in the revised manuscript.
>
> **Reply to W4 & Q2**
>
> We thank the reviewer for this question. Our experimental setup follows the evaluation framework adopted in prior works such as APN and HyperIMTS. Specifically, the configurations and hyperparameters of baseline models are inherited from this widely recognized framework, which provides well-established and carefully tuned implementations for fair comparison. By using the officially released scripts and default settings within this framework, we ensure that all baselines are evaluated under consistent and optimized conditions, thereby maintaining the fairness and reliability of the comparison.

---

> > ### Author Rebuttal · Reviewer_y91f · 2026-04-03
> >
> > The authors’ response addresses all my concerns. Since my initial score was already high, I keep the original rating unchanged and remain supportive of acceptance.

---

> > > ### Author Response · Authors · 2026-04-04
> > >
> > > Dear Reviewer, we are thrilled that our responses have effectively addressed your questions and comments. We would like to express our sincerest gratitude for taking the time to review our paper and provide us with such detailed feedback!

---

### Official Review · Reviewer_AK9E · 2026-03-12

**Soundness:** 3
**Presentation:** 3
**Significance:** 3
**Originality:** 2
**Overall Recommendation:** 4
**Confidence:** 4

**Summary:**

The paper proposes TFMixer a jointed time frequency modeling for irregular multivariate time series forecasting. The model consists of Global Frequency Module and Local Time Module in parallel. It uses a learnable Non Uniform Discrete Fourier Transform to capture global, while using query based patch mixing to process local information. Experiments on several real world dataset show improvement over prior methods.

**Compliance With Llm Reviewing Policy:**

Affirmed.

**Final Justification:**

The authors have addressed most of my concerns. However, one of the main design choices of the proposed method is to learn frequencies instead of using fixed frequencies such as DFT. It would be very beneficial for the paper to include a short ablation study comparing the learned frequencies with the frequencies of the dataset. There are two possible scenarios: if the learned frequencies align with the dataset’s frequency distribution, this could suggest that the model effectively keep important information; on the other hand, it may also help us better understand the structure of the dataset. Either way, this would improve the interpretability and usefulness of the model. I hope the authors can include this in the final version of the paper, since the rebuttal timeline may be too short to conduct these experiments.

I will keep my score as the author was not able to provide this during the rebuttal phase.

**Key Questions For Authors:**

1. What is the computational overhead of the NUDFT module compared to standard spectral approaches?
2. How sensitive is TFMixer to different irregularity levels and missing data patterns?
3. What is the model size compare to others?

**Limitations:**

The author did not provide a limitation section. The author could discuss robustness of extreme sparsity or long sequence length, or lack of theoretical guarantee.

**Strengths And Weaknesses:**

Strengths:

Soundness:
The design of separate modules for global and local dynamics is standard in time series forecasting. The use of learnable NUDFT is technically reasonable.

Presentation:
The paper is clearly motivated by the irregular time series problem. Each component is clearly explained with detailed notations.

Significance: Irregular multivariate time series forecasting is important in various domains, making the problem practically relevant. Future works that utilize DFT could use the NUDFT as a frequency decomposition module.

Originality: The integration of learnable NUDFT with local patch based temporal mixing provide a novel hybrid frame work.

Weaknesses:

Soundness:
The paper provides limited theoretical justification for the learnable NUDFT formulation. Also there is a lack of ablation study of learnable $\omega$, this hinders the explainability of model selecting certain $\omega$ for a given dataset.


Significance: While improvements are reported, the empirical gain over strong baseline appears moderate, sometimes worse than the baseline, in particular PhysioNet dataset.

Originality:  Several components build on existing ideas, and the novelty mainly lies in their combination.

---

> ### Author Rebuttal · Authors · 2026-03-30
>
> Dear Reviewer AK9E, we sincerely thank you for the valuable feedback.
>
> **Response to Concerns on Soundness**
>
> The design of TFMixer is grounded in spectral analysis theory. Specifically, it extends the traditional Discrete Fourier Transform (DFT) to a learnable Non-Uniform Discrete Fourier Transform (NUDFT), where observations are projected onto non-orthogonal basis functions of the form $ \exp(-j 2\pi \omega_k t_l) $. This formulation naturally addresses spectral distortion caused by irregular timestamps from a theoretical perspective.
>
> Regarding the learnability of $ \omega $, we note that the ablation results in Table 2 already provide supporting evidence. In particular, removing the Global Frequency Module (w/o Spectrum Refinement) leads to a clear performance drop, and further removing the spectrum refinement mechanism also degrades performance. These results indicate the importance of learnable frequency components.
>
> **Response to Concerns on Significance**
>
> While the performance gains on individual datasets may vary, TFMixer demonstrates strong and consistent generalization across diverse domains. Specifically, it ranks first in the majority of cases across benchmarks spanning four different areas, including healthcare and climate data. This consistency indicates that the proposed method is not tailored to any specific dataset, but achieves robust performance across heterogeneous scenarios. Regarding the PhysioNet dataset, although TFMixer does not achieve the best performance in terms of MSE, it still attains the best results under the MAE metric.
>
> **Response to Concerns on Originality**
>
> We would like to clarify that the novelty of our work does not lie in a simple combination of existing ideas. Specifically, we propose a unified joint time–frequency modeling framework for irregular multivariate time series forecasting, which explicitly bridges global spectral dependencies and local temporal dynamics—an aspect that has not been adequately addressed in prior work. In addition, the proposed learnable NUDFT constitutes a key methodological contribution. Unlike traditional DFT-based approaches, it introduces a learnable non-uniform spectral representation tailored to irregular timestamps, enabling more accurate and flexible frequency modeling in real-world scenarios.
>
>
> **Reply to Q1**
>
> The computational overhead of the proposed NUDFT module mainly arises from the explicit projection onto sinusoidal bases with learnable frequencies. Specifically, given an input sequence of length $L$ and $K$ learned frequencies, the computational complexity is **$O(L \cdot K)$**, as it involves evaluating sine/cosine bases and performing weighted summation over valid timestamps.
>
> In contrast, standard FFT-based methods have a complexity of **$O(L \log L)$**, but they rely on the uniform sampling assumption and cannot be directly applied to irregular time series. In practice, they typically require interpolation or imputation as preprocessing steps, which introduce additional computational overhead and may lead to information distortion.
>
> In addition, the subsequent spectrum encoding (MLP) operates on a $2K$-dimensional representation and has a complexity of **$O(K^2 + K \cdot d)$**. Since this computation is independent of the sequence length $L$, and $K$ is relatively small, its overhead is well controlled and contributes only a minor portion of the overall computation.
>
>
> **Reply to Q2 & Limitations**
>
> We evaluate TFMixer on multiple widely used datasets that exhibit diverse irregularity levels and missing data patterns, spanning domains such as healthcare, biomechanics, and climate science. Specifically, these datasets vary substantially in terms of sample size (5,400–26,736), number of variables (5–96), and average sequence length (46–163), while their missing ratios range from 75% to 96.7%. This diversity constructs a comprehensive evaluation setting that covers varying levels of data complexity and irregularity, enabling a systematic assessment of the model’s robustness across diverse real-world scenarios. To further evaluate the robustness of the model, we have already provided additional experiments in Appendix B.2 (Varying Lookback Lengths and Forecast Horizons) in the original paper. The results show that as the lookback window increases—corresponding to higher irregularity and more severe missingness—TFMixer consistently outperforms strong baselines such as tPatchGNN, demonstrating its effectiveness under large-scale missing data and highly asynchronous conditions. Moreover, across longer forecasting horizons, TFMixer continues to outperform baselines, further confirming its robustness under diverse temporal settings.
>
> **Reply to Q3**
>
> We have supplemented the paper with additional efficiency experiments to provide a clearer comparison of model size and computational cost. Please refer to the results at the **following link**: https://anonymous.4open.science/r/ICML-TFMixer-Rebutal-0863/model_size.md

---

> > ### Author Rebuttal · Reviewer_AK9E · 2026-04-03
> >
> > Thank you for the response. It addresses most of my concerns. Since the model has a learn frequency component, it would be beneficial to demonstrate with example on one of the dataset that the learn frequency align with what we expect with the characteristic of the dataset. This provides confidence that the learn frequency is useful.
> > I will keep my current score. I will be happy to increase the score if the author address my concern.

---

> > > ### Author Response · Authors · 2026-04-04
> > >
> > > Dear Reviewer AK9E, thank you for your positive feedback and constructive suggestion.
> > >
> > > To verify that the learned frequency component effectively captures periodic patterns, we provide a qualitative visualization on the Human Activity dataset. The results are available at:
> > > https://anonymous.4open.science/r/ICML-TFMixer-Rebutal-0863/visualization.md
> > >
> > > As shown in the visualization, compared with APN, Hi-Patch, and KAFNet, our TFMixer better captures the inherent periodic characteristics of the dataset. This provides direct evidence that the learned frequency components align well with the underlying temporal structure.
> > >
> > > In addition, the ablation results in Table 2 further demonstrate the effectiveness of the frequency module, where removing it consistently degrades performance.
> > >
> > > We sincerely appreciate your time and valuable feedback, which we believe helps improve the clarity and completeness of our work.

---

### Decision · Program_Chairs · 2026-04-30

**Decision:**

Accept (regular)

**Comment:**

This paper addresses an important and practically relevant problem in irregular multivariate time series forecasting with a well-motivated hybrid design that combines global frequency modeling and local temporal modeling.  The proposed method is technically sound overall, and the use of a learnable NUDFT together with local patch-based mixing offers a meaningful and reasonably novel contribution beyond standard architectures. The empirical results are generally strong across multiple real-world datasets, and the rebuttal addressed most of the major concerns, although additional analysis of the learned frequencies would further improve interpretability. Overall, the paper makes a solid contribution to the area and merits acceptance.